# A local interplay between diffusion and intraflagellar transport distributes TRPV-channel OCR-2 along *C. elegans* chemosensory cilia

Jaap van Krugten [1], Noémie Danné[1] & Erwin J. G. Peterman [1✉]

To survive, *Caenorhabditis elegans* depends on sensing soluble chemicals with transmembrane proteins (TPs) in the cilia of its chemosensory neurons. Cilia rely on intraflagellar transport (IFT) to facilitate the distribution of cargo, such as TPs, along the ciliary axoneme. Here, we use fluorescence imaging of living worms and perform single-molecule tracking experiments to elucidate the dynamics underlying the ciliary distribution of the sensory TP OCR-2. Quantitative analysis reveals that the ciliary distribution of OCR-2 depends on an intricate interplay between transport modes that depends on the specific location in the cilium: in dendrite and transition zone, directed transport is predominant. Along the cilium motion is mostly due to normal diffusion together with a small fraction of directed transport, while at the ciliary tip subdiffusion dominates. These insights in the role of IFT and diffusion in ciliary dynamics contribute to a deeper understanding of ciliary signal transduction and chemosensing.

[1] LaserLaB and Department of Physics and Astronomy, Vrije Universiteit Amsterdam, De Boelelaan, 1081 HV Amsterdam, The Netherlands.
✉email: e.j.g.peterman@vu.nl

Organisms depend on avoidance behavior to translocate themselves to a more favorable environment[1–5]. In most cases, the ability to sense the outside environment is dependent on transmembrane proteins (TPs) triggering and amplifying downstream signaling pathways, including neuronal activity, that evoke a behavioral response. Primary cilia are slender protrusions of eukaryotic cells that function as signaling hubs, harboring a myriad of TPs involved in signal transduction[6–9]. In general, external stimuli are sensed by G-protein coupled receptors (GPCRs), which relay signals via a series of second messenger molecules like cAMP and calcium to elicit a behavioral or genomic response[10,11]. The ciliary distribution of these TPs is essential for their function, and their location is regulated by the intraflagellar transport (IFT) machinery[12,13]. Previous work on the dynamics of GPCRs in mammalian cell lines has shown that they mostly move by passive diffusion, although active transport is observed, most notably during signal-dependent retrieval of GPCRs out of the cilia[14].

The nematode *C. elegans* has proven to be an invaluable model organism to study proteins involved in signal transduction and has been widely used as a model for avoidance behavior and the dynamics of ciliary TPs[15–18]. *C. elegans* sensory cilia are located at the end of dendrites of chemosensory neurons in the amphid and phasmid channels, situated in the head and tail regions of the worm, respectively. The development, maintenance, and function of these cilia critically depend upon a specific transport mechanism, IFT. IFT consists of transport units called IFT trains, that travel from the ciliary base to the tip and back again, facilitating the transport of ciliary building blocks and sensory proteins[19]. In *C. elegans* cilia, anterograde transport (from base to tip) is driven by heterotrimeric kinesin-II in the first few micrometers of the cilium, after which it gradually hands over the IFT trains to homodimeric kinesin-2 OSM-3[20,21]. At the tip of the cilium, IFT trains disassemble and remodel[22]. Retrograde transport, back to the base, is driven solely by IFT dynein (cytoplasmic dynein-2)[23–25].

The backbone of these trains is formed by the IFT-A, IFT-B and BBSome particle complexes. Studies in mouse, *Chlamydomonas*, and *C. elegans* demonstrated that TPs such as GPCRs and the ion-channel OCR-2 are, besides exhibiting diffusive motion, an IFT cargo[26–33]. The link between IFT trains and functional cargo is most likely formed by Tubby proteins and the BBSome. Tubby-like protein 3 (TULP3), together with IFT-A, was found to be essential for the ciliary import of some GPCRs[13,34]. Mutations in the BBSome and the *C. elegans* TULP3 homolog, tub-1, show defects in chemotaxis behavior, demonstrating the importance of the localization and active transport of sensory TPs[12,13]. It has recently been shown in *C. elegans* phasmid chemosensory cilia, that OCR-2 redistributes over the cilium after stimulation with adversive chemicals (SDS and $Cu^{2+}$) or hyperosmotic solutions (glycerol and high concentrations of NaCl)[35].

Although transport mechanisms of TPs in different cilia appear similar, there are large architectural differences between *C. elegans* sensory cilia, mammalian primary cilia, and *Chlamydomonas* flagella[36]. *C. elegans* cilia are ~7 μm long, with microtubule (MT) doublets occupying only about half the ciliary length and MT singlets extending from there to the ciliary tip. In contrast, *Chlamydomonas* flagella are ~10–14 μm long and mammalian primary cilia ~4–5 μm. Both contain MT doublets along almost their entire length. Furthermore, whilst mammalian primary cilia and *Chlamydomonas* flagella are exposed to their environment over their entire length, *C. elegans* sensory cilia reside inside a channel, with only the ciliary tip close to the channel opening towards the environment.

Despite progress in understanding ciliary architecture and transport of TPs in cilia, many questions remain. It is, for example, unclear what the interplay is between active, IFT-driven, and passive, diffusive transport of TPs, and whether these motility modes are location specific. Here, we perform sensitive, fluorescence microscopy in the sensory cilia of live *C. elegans* to reveal the single-molecule dynamics of a functional IFT cargo protein, the TRPV calcium channel OCR-2. Analysis of the trajectories reveals that active transport is the predominant mode of transport in the dendrite and transition zone. Along the cilia, the proper distribution of OCR-2 depends on an intricate location-specific interplay between active transport, normal diffusion, and subdiffusion, with the latter playing an important role in the confinement of TPs at the ciliary tip. These insights into the location-specific motility modes of a functional cargo of IFT contribute to a wider understanding of IFT dynamics and to cilia as chemosensory signal transducers.

## Results

The nervous system of *C. elegans* consists of 302 neurons, of which 32 are ciliated. Most of these cilia are situated in the two amphid openings, in the head region of the nematode. Here, we study the two pairs of phasmid cilia (PHAL, PHAR, PHBL, and PHBR), located in the tail of the animal. To this end, we generated worms endogenously expressing the fluorescently labeled GPCR SRB-6 (SRB-6::EGFP) and the transmembrane calcium channel OCR-2 (OCR-2::EGFP)[37,38]. Expression levels of SRB-6::EGFP were low, and only several tens of molecules could be observed in each phasmid cilia pair (Supplementary Fig. 1 and Supplementary Movie 1). Such expression levels are too low to perform single-molecule experiments. Expression levels of OCR-2, on the other hand, were much higher, making it possible to perform live single-molecule imaging experiments.

**Ensemble distribution and dynamics of OCR-2**. First, we investigated the ensemble distribution and dynamics of OCR-2. To visualize the steady-state distribution of OCR-2 along the cilium, time-averaged fluorescence images were generated from image sequences of a *C. elegans* strain expressing OCR-2::EGFP (Fig. 1a–d). In order to localize the TZ with respect to the OCR-2 distribution, we imaged phasmid cilia in worms overexpressing the TZ marker MKS-6::mScarlet and with endogenously labeled OCR-2 (Fig. 1c, d). From such images, the intensity distribution along the ciliary long axis was calculated (Fig. 1e). The images and intensity distribution indicate that the TZ corresponds to a region with very low OCR-2 density. Anterior to the TZ, OCR-2 appears to accumulate in a cup-like shape, which most likely corresponds to the Periciliary Membrane Compartment (PCMC). Posterior to the TZ, the intensity of OCR-2 is relatively high, dropping further along the middle segment and increasing again in the distal segment (DS), peaking at the ciliary tip. A super-resolution image obtained from many localizations of individual OCR-2 molecules (see below and ref.[39]), confirms these observations (Fig. 1b), highlighting the membrane localization of OCR-2, particularly in the wide proximal segment (PS).

To obtain a first insight into what underlies the distribution of OCR-2 along the cilium, we generated kymographs from the original image sequences at the ensemble level (Fig. 1f). The kymograph clearly shows the directional, processive movement of OCR-2, in agreement with previous studies of worms expressing fluorescent OCR-2 from extra-chromosomal arrays[30]. The continuous lines resulting from motor-driven OCR-2, however, are less clear than what we have observed before for IFT-train components[20,24] and appear to be obscured by a relatively high, unstructured background signal of OCR-2. This might indicate that OCR-2 is only partly actively transported by IFT and to a substantial account by passive diffusion.

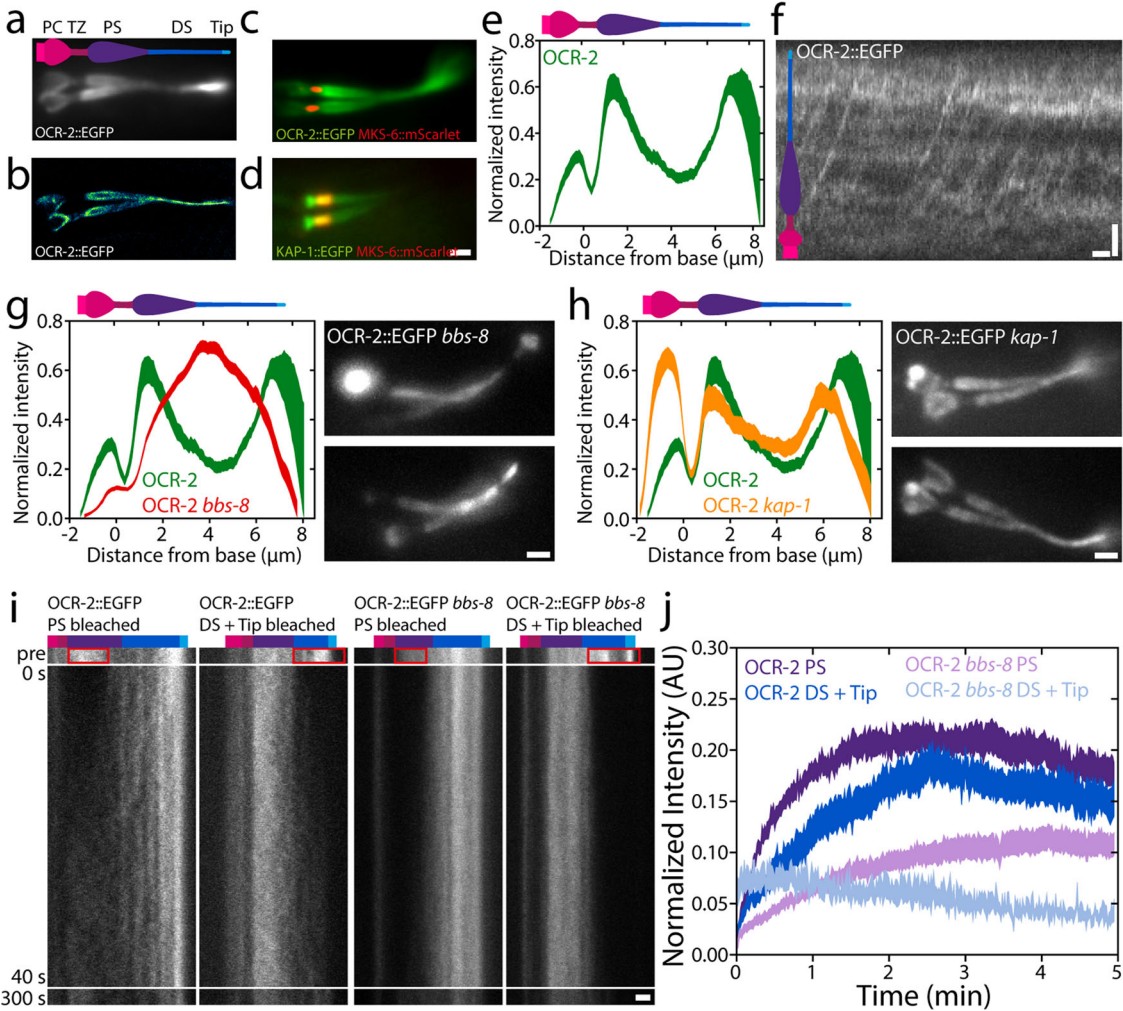

**Fig. 1 Ensemble distribution and dynamics of OCR-2. a** Time-averaged (over 3 s) fluorescence images of OCR-2::EGFP. **b** Super-resolution image of OCR-2::EGFP molecule localizations. **c** Dual-color image of endogenously labeled OCR-2::EGFP and overexpressed MKS-6::mScarlet. **d** Dual-color image of endogenously labeled Kinesin-II(KAP-1::EGFP) and MKS-6::mCherry. Scale bar: 1 μm. **e** Average intensity distribution of OCR-2::EGFP along the cilium ($n = 22$ cilia, averaged normalized intensity distributions). **f** Kymograph of ensemble dynamics of OCR-2::EGFP with bars indicating the PS (purple) and DS (dark blue). Scale bars: 1 μm and 1 s. **g** Left: averaged normalized average intensity distribution of OCR-2::EGFP (green) and OCR-2::EGFP *bbs-8* (red) ($n = 71$) along the cilium. Right: Time-averaged fluorescence images of OCR-2::EGFP *bbs-8*. Scale bar: 1 μm. **h** Left: normalized average ciliary distribution of OCR-2::EGFP (green; same data as **e**) and OCR-2::EGFP, kap-1 mutant (red) ($n = 26$ worms). Right: Time-averaged (over 3 s) fluorescence images of OCR-2::EGFP, kap-1 mutant. Scale bar: 1 μm. **i** Representative kymographs before and after photobleaching the PS, or DS and Tip of OCR-2::EGFP and OCR-2::EGFP *bbs-8*. Colored bars indicate segments as in **f**. Time scale is identical for all kymographs. Scale bar: 1 μm. **j** Averaged normalized intensity of OCR-2::EGFP in PS, or DS and Tip, after photobleaching PS or DS and Tip respectively, in WT and *bbs-8* animals (OCR-2::EGFP PS: $n = 11$, DS = Tip: $n = 10$; OCR-2::EGFP *bbs-8* PS: $n = 16$, DS + Tip: $n = 10$, linewidth represents s.e.m.).

To determine how essential active transport by IFT is for the ciliary distribution of OCR-2, we generated worms lacking BBS-8, which is crucial for a functional BBSome and thereby for the connection between ciliary membrane proteins and the IFT machinery[12]. Kymographs generated from image sequences, indeed, do not show any signs of active transport of fluorescent OCR-2 in the absence of BBS-8 function, but do show that IFT is still active (Supplementary Figure 2). Fluorescence images were time averaged to generate an intensity distribution averaged over multiple cilia (Fig. 1g). These images show that OCR-2 does enter the cilia of *bbs-8* worms, but that its distribution along the cilium is remarkably different compared to wild-type animals: OCR-2 concentration does not peak behind the TZ and at the tip (as found in wild type) but shows a maximum in the middle of the cilium. We note that the time-averaged fluorescence images of *bbs-8* worms were quite heterogeneous. In some worms, substantial accumulations of OCR-2 were observed in the PCMC

(e.g. upper image in Fig. 1g) suggesting that the import of OCR-2 is severely hampered in these cilia. Together, these results show that active transport plays an important role in the ciliary distribution of the TP OCR-2. Furthermore, although a functional BBSome has been shown to be necessary for avoidance behavior and is involved in the ciliary exit of TPs[12,40], it seems not completely essential for ciliary entry of OCR-2 into the cilium. Together, this might indicate that the proper, IFT-dependent, ciliary distribution of TPs involved in chemotaxis is required for the propagation of environmental stimuli.

To study the effect of a more subtle modulation of TP active transport, we generated worms with endogenously labeled OCR-2, but lacking Kinesin-II function (ok676). Kinesin-II drives, together with OSM-3, anterograde IFT, with Kinesin-II mostly active from the ciliary base across the TZ and OSM-3 in the rest of the cilium[20]. While mutant worms lacking OSM-3 function have short cilia lacking a DS and are defective in osmotic

avoidance, worms lacking Kinesin-II function do show osmotic avoidance behavior and exhibit only minor structural defects[39,41]. Time-averaged image sequences and the average intensity distribution of OCR-2 in kap-1 mutant cilia show accumulations of OCR-2 posterior to the TZ and at the ciliary tip, just as wild type, but also show a far more pronounced accumulation in the PCMC (Fig. 1h). These observations are in line with earlier observations of the distributions of IFT-particle components in kap-1 mutant strains[20]. In this study, not only similar ciliary distributions of IFT components were observed in the mutant strains as in wild type but also show accumulations in the PCMC, suggesting that Kinesin-II is a more efficient import motor than OSM-3. Furthermore, our observations of OCR-2 in OSM-3 mutant worms are in line with those in bbs-8 mutant strains, supporting the conclusion that efficient, active transport of OCR-2 by IFT is essential for its distribution and function.

To obtain insight into the ensemble dynamics that underlie the ciliary distribution of OCR-2, we performed Fluorescence Recovery After Photobleaching (FRAP) experiments on OCR-2::EGFP in wild-type and bbs-8 mutant worms (Fig. 1i), photobleaching the EGFP in PS, or DS and tip, by brief, intense laser illumination. Image sequences of the fluorescence recovery were converted into kymographs of the cilium (Fig. 1i) and fluorescence intensity time traces integrated over the ciliary segments (Fig. 1j). Kymographs and time traces show that, in wild-type cilia, the rate and extent of OCR-2::EGFP recovery in the PS after bleaching the PS, is slightly different from the recovery in DS and Tip after bleaching DS and Tip. In the PS, only ~21% of the intensity before bleaching the PS recovered after ~2 min, while in the DS and Tip ~18% recovered ~2 min after bleaching DS and Tip. We note that, given the experiment uncertainty (e.g. photobleaching during the experiment) the data can only be interpreted qualitatively, and differences in the mobile fraction of several percent are not statistically significant. For a detailed quantitative analysis, we performed single-molecule experiments (see below). Our FRAP data qualitatively indicate that in the cilium, only a small fraction of OCR-2 is free to move, driven by IFT or free diffusion.

In bbs-8 mutant worms (where the connection between OCR-2 and IFT machinery is disrupted), the mobile fraction appears to be even smaller in the PS (~11%) and DS (~5%) compared to those segments in wild type. Compared to wild type, a similar difference in the recovery rate and extend between the PS and DS in the bbs-8 mutant cilia was observed. The difference of the mobile fractions in the PS and DS between wild-type and bbs-8 mutant worms furthermore demonstrates the importance of IFT for the ciliary distribution of OCR-2. This difference is larger in the DS, suggesting IFT plays a larger role in the DS for OCR-2 localization.

Collectively, these findings demonstrate the diversity of transport modalities underlying the ensemble dynamics resulting in the steady-state ciliary distribution of OCR-2. This distribution is not uniform along the cilium but shows accumulations at the Tip and posterior to the TZ, and a smaller accumulation anterior to the TZ. In mutant animals with perturbed active transport of OCR-2, the distribution is different. In addition, our imaging and FRAP analysis suggests that diffusion of OCR-2 in the ciliary membrane also plays a key role.

**Single-molecule imaging reveals large diversity in OCR-2 motility**. To obtain more detailed and quantitative insight into the extent by which diffusion and active transport drive OCR-2 motility along the cilium, we performed single-molecule tracking of OCR-2::EGFP in live *C. elegans* using laser-illuminated wide-field epifluorescence microscopy, at a frame rate of 10 Hz. In Fig. 2a (see also Supplementary Movie 2), example trajectories are represented as kymographs, with a location in the cilium indicated. The kymographs suggest a variety of different behaviors of OCR-2, ranging from diffusion to directed transport in anterograde and retrograde directions. The specific behavior appears to depend on the actual location in the cilium. In the dendrite, straight, slanted lines are observed towards the PCMC (see also Supplementary Fig. 3) (1), indicative of active, directional transport of OCR-2. In the PCMC, kymograph lines appear mostly horizontal but slightly wavy (2), indicative of particles lingering for a substantial amount of time (which appears to be limited by photobleaching in our experiments). In the TZ, again straight, slanted lines are observed (3), indicating that particles cross the TZ mostly by directed transport. In the PS, more heterogeneous behavior is observed. Some particles show straight, slanted lines (4), indicative of active transport, while others appear saltatory (5), indicating that particles diffuse, or move by short bouts of active transport in an anterograde or retrograde direction. In the DS, a similar pattern of directed motion in combination with saltatory movement is observed (6), albeit the fraction of directed transport appears higher. Taken together, single-molecule kymographs hint at a large diversity of OCR-2-motility patterns that appear to depend on the specific ciliary segment and location.

One of the observations in the kymographs of Fig. 2a is that the motion of OCR-2 in the TZ appears to be mostly directional, driven by IFT. The motility of cargo OCR-2 appears very similar to that of motor Kinesin-II (imaged using a KAP-1::EGFP strain; Fig. 2b), with comparable velocity in the retrograde direction and slightly higher in anterograde direction (Fig. 2c). These observations strongly suggest that OCR-2 is indeed transported across the TZ by IFT trains driven by Kinesin-II in the anterograde direction and IFT dynein in the retrograde direction and that crossing the TZ by diffusion does not occur. This is consistent with the earlier observation that the densely packed protein network (e.g., the Y-shaped linkers connecting axoneme and membrane) of the TZ forms a diffusion barrier between dendrite and PS[42]. We have shown before that in the absence of Kinesin-II function (in the kap-1 mutant strain), OSM-3 can take over and drive IFT across the TZ. In these strains, however, the frequency and number of IFT trains entering the PS appear affected and IFT components show more substantial accumulations in front of the TZ[20]. In line with these observations, we have shown above that OCR-2 is still present in cilia lacking kinesin-II function, but that its ciliary distribution is affected (Fig. 1h). Single-molecule kymographs generated from image sequences of kap-1 mutant worms show many more OCR-2 molecules being stuck in the TZ (Fig. 2b) or traversing it more irregularly than in wild-type animals, although successful TZ crossings can still be observed. Velocities extracted from the trajectories were only slightly lower than for wild type (Fig. 2c), which might seem remarkable, since OSM-3, which is generally considered to be a faster motor, drives anterograde crossing (in the kap-1 mutant) instead of Kinesin-II (in wild type). We have, however, shown before that OSM-3 velocity in the TZ of kap-1 mutant worms is substantially reduced in addition to affecting OSM-3's efficiency to traverse the TZ, most likely due to TZ structures acting as roadblocks for OSM-3, more severely than for Kinesin-II[20]. Taken together, the TZ acts as an efficient diffusion barrier for OCR-2, which needs to be overcome by active transport driven by IFT. In mutant worms lacking Kinesin-II function, OCR-2 transport over the TZ can be taken over by OSM-3, albeit with a loss of efficiency. In both wild-type and kap-1 mutant worms, IFT dynein drives effective retrograde transport of OCR-2.

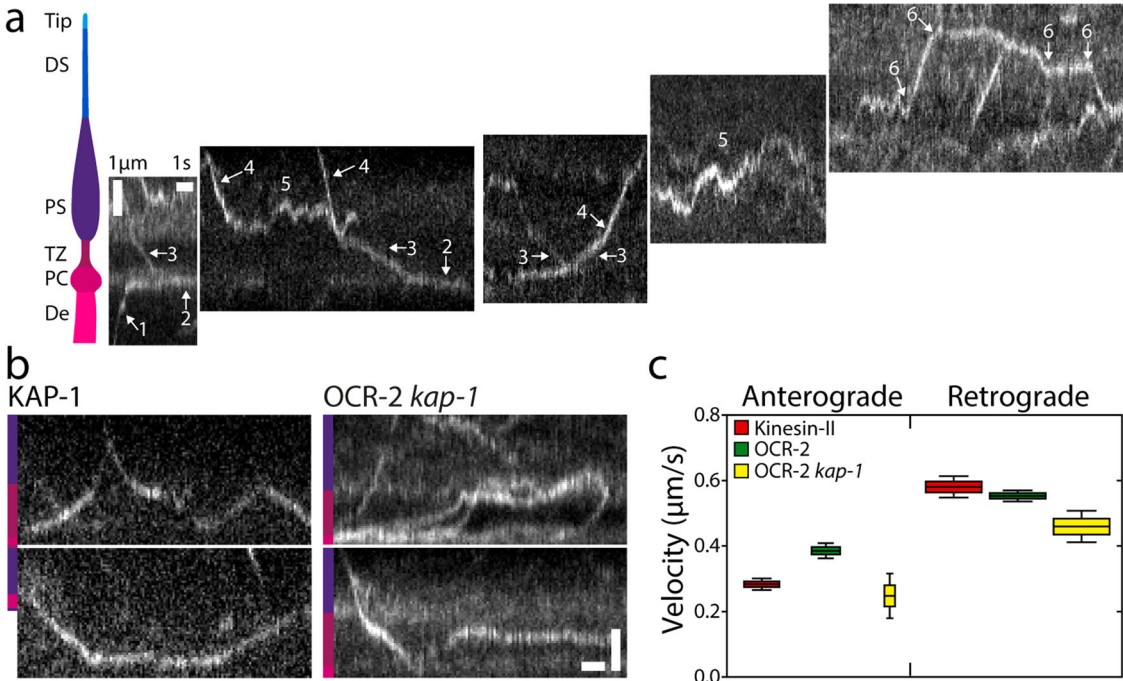

**Fig. 2 Single-molecule imaging of OCR-2. a** Cartoon of the ciliary segments and representative single-molecule kymographs of OCR-2::EGFP. Scale bars, vertical: 1 μm, horizontal: 1 s. **b** Single-molecule kymographs showing TZ crossings of Kinesin-II (KAP-1::EGFP) and OCR-2::EGFP *kap-1*(ok676). Color bars indicate ciliary segment as in (A). Scale bars, vertical: 1 μm; horizontal: 1 s. **c** Anterograde (A) and retrograde (R) velocity of KAP-1::EGFP (A: 63 trajectories; R: 273 trajectories), OCR-2::EGFP (A: 51 trajectories; R: 297 trajectories) and OCR-2::EGFP *kap-1*(ok676) (A: 19 trajectories; R 141 trajectories). Velocities were extracted using KymographDirect from kymographs optimized for the TZ (a different data set than used in the rest of the manuscript). Data are represented as boxplot. Indicated are median (block horizontal line), 25th–75th percentile (height of the colored boxes), 5th–95th percentile. The widths of the boxes represent the sample size.

**Single-molecule imaging reveals bimodal motility distribution of ciliary OCR-2**. To get a more quantitative insight into the motility behavior of single OCR-2 molecules inside the entire cilium, we tracked single molecules in images sequences, such as those represented in the kymographs of Fig. 2a, using a local ciliary coordinate system (parallel and perpendicular to a spline drawn along the long axis of each cilium). Representative trajectories are shown in Fig. 3a, with examples of a trajectory suggestive of active, directed transport (*), a trajectory suggestive of diffusive transport (+), and a trajectory of a molecule that hardly moves in the ciliary tip (#). For further analysis, we calculated the mean squared displacement (MSD) of these example trajectories (Fig. 3b, c). MSD analysis is a widely used approach to characterize motility, based on the scaling of MSD with time[43]. This scaling can be expressed in one dimension as:

$$\mathrm{MSD}(\tau) = 2 \cdot \Gamma \cdot \tau^{\alpha} \qquad (1)$$

with Γ the generalized transport coefficient, τ the time lag, and α the exponent. In case α = 2, the motion is purely directed (ballistic) and the velocity (v) is equal to $\sqrt{2\Gamma}$. When α = 1, the motion is due to normal diffusion and is characterized by a diffusion coefficient (D) equal to Γ. In case α < 1, the motion is considered subdiffusive, which in this case could be caused by the association of OCR-2 to stationary structures and/or by the restriction of OCR-2 motion due to other, slower moving proteins[44]. Application of this MSD analysis to the three example trajectories confirms the different modes of motility underlying these trajectories: the MSD of the apparently directed trajectory (*) scales roughly quadratically with time, that of the diffusive trajectory (+) linearly, and that of the hardly moving molecule (#) with a slope less than linear, indicative of subdiffusion.

The analysis used above was applied to user-selected (fragments of) trajectories with different lengths. To obtain unbiased insight from all 185 trajectories (in 23 worms) constituting 7237 individual localizations, we calculated the instantaneous MSD, using a sliding window of 15 consecutive locations, in accordance with previous work[45], yielding instantaneous values of Γ and α, both parallel and perpendicular to the cilium (Fig. 3d and "Methods")[46]. As a first characterization of the data obtained in this way, we plotted all $\alpha_{//}$ values in a histogram and corresponding violin plot (Fig. 3e). Histogram and violin plot clearly show a bimodal distribution of $\alpha_{//}$ confirming that OCR-2 moves by a combination of directed and diffusive motion. A fit with two Gaussians to the histograms yields a diffusive fraction (84%) with an average $\alpha_{//}$ of 0.66 (Full Width Half Maximum (FWHM): 0.56), and a directed motion fraction (16%) with an average $\alpha_{//}$ of 1.78 (FWHM: 0.17). This indicates that most of the motion of OCR-2 in the cilia is diffusive. Furthermore, the diffusive fraction is broadly distributed, indicating heterogeneous diffusive behavior. In the remainder of this study, we will use α = 1.4, the intersection between the two Gaussians fits, as the threshold to discriminate between diffusive and directed motion. This analysis indicates that values for the instantaneous diffusion coefficient can be readily obtained from single-molecule trajectories of OCR-2 in cilia and can provide quantitative insight in OCR-2 dynamics.

**A location-specific interplay of transport dynamics underlies the ciliary distribution of OCR-2**. Next, we used the same dataset to obtain more quantitative insight into the motion of OCR-2 in the different ciliary segments. To obtain an overview of the location-specific motility properties of OCR-2, color coded $\alpha_{//,\perp}$ and $D_{//}$ values, derived from the instantaneous MSD, were plotted in

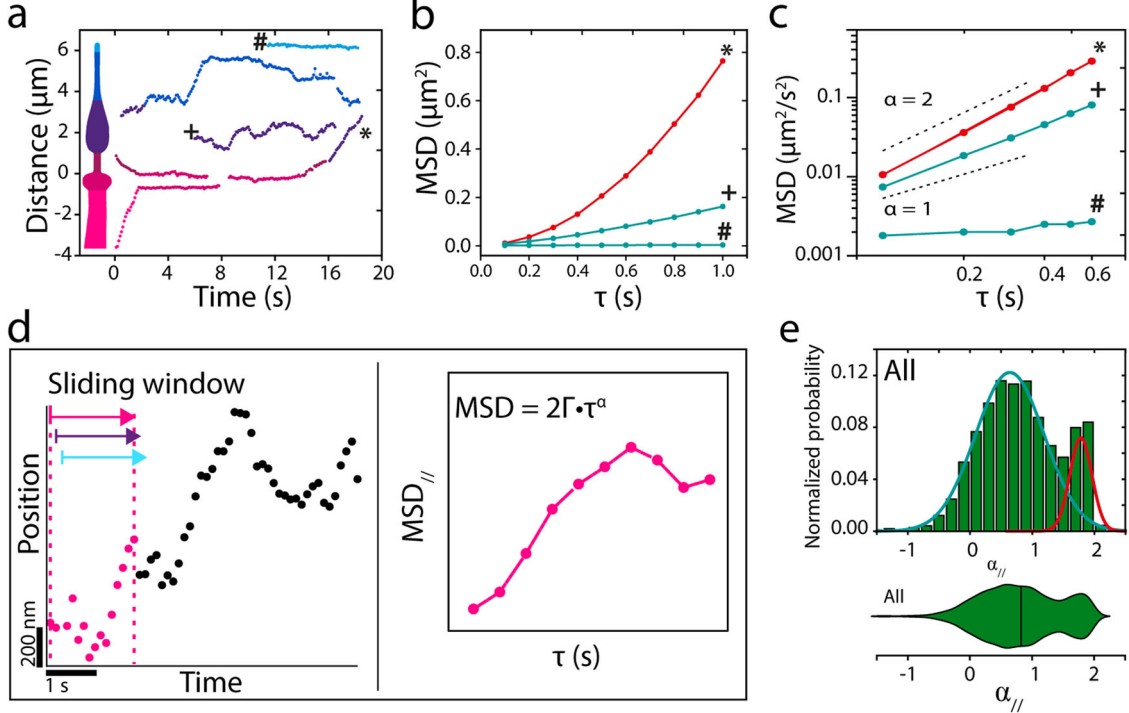

**Fig. 3 Analysis of single-molecule OCR-2 trajectories. a** Example single-particle trajectories from the kymographs of Fig. 2a. **b** Mean squared displacement (MSD) of annotated trajectories in **a**, demonstrating active transport (*), normal diffusion (+), and subdiffusion (#). **c** Log–log plot of the first six time points. **d** Cartoon explaining the single-molecule data analysis with a sliding window. **e** Histogram and corresponding violin plot of normalized probability of $\alpha_{//}$ for all 7237 data points obtained from 23 nematodes. Colored lines represent two underlying Gaussian distributions.

function of their location (Fig. 4a). Even though the data used for these heat maps is derived from 23 different animals, the resulting shape resembles the shape of a single cilium, validating the robustness of our imaging and analysis. This representation furthermore highlights that OCR-2 motility parameters depend on the location in the cilium. To further quantify this, we represented the distributions of the $\alpha_{//}$ values in the different ciliary segments in violin plots (Fig. 4b). The violin plots show that the distributions vary substantially between the segments. In the TZ, $\alpha_{//}$ values peak close to two; in the PS $\alpha_{//}$ values are widely distributed, peaking around 1; in the DS, two peaks can be observed, the major, broad one peaking around 0.5 and the minor one peaking slightly below 2; in the Tip, a single distribution around 0.5 can be seen. The results of Gaussian fits to these distributions (Supplementary Fig. 4) are represented in Supplementary Table 1, which shows the fractional contributions for directed motion and diffusion in the different ciliary compartments and the corresponding average $\alpha_{//}$ values. Taken together, Fig. 4b and Supplementary Table 1 confirm, in a quantitative way, the initial indications obtained from the kymographs. In the TZ, OCR-2 moves mostly by directed transport; in the PS, OCR-2 moves mostly by normal diffusion in the membrane and only to a small extent via directed transport; in the DS, the fraction of OCR-2 moving by directed transport is slightly higher, while the rest moves subdiffusively; at the tip all motion is subdiffusive.

We further analyzed our data by determining the apparent local velocities from (stretches of the) trajectories with $\alpha_{//} \geq 1.4$ using:

$$v_{app} = (x(t + dt) - x(t))/dt \quad (2)$$

with d$t$ the frame integration time[47], averaging locally over 10 displacements (Fig. 4c). Local apparent anterograde velocity compares very well to velocity profiles obtained before using bulk[20] and single-molecule fluorescence imaging[39] of kinesin

motors and other IFT components: with a velocity of about 0.5 μm/s close to the TZ, gradually increasing in PS, reaching the maximal value of more than 1 μm/s. This similarity of the anterograde OCR-2 velocity profiles presented here, compared to those obtained from the IFT components reported before, highlights the validity of our approach. Apparent retrograde velocities, however, agree less well with velocities reported before and are lower, in particular in the DS. We do not have a definitive explanation for this, but we note that the number of events showing directed minus end motion is substantially lower than for anterograde, potentially indicating that bouts of retrograde transport of OCR-2 are only short lived (compared to our sliding window of 15 frames, corresponding to 1.5 s). In agreement with this, the OCR-2 kymographs show only few straight retrograde events in contrast to anterograde events, which are far more evident.

We next looked into the values of the apparent diffusion coefficients parallel and perpendicular to the ciliary axis ($D_{//,\perp}$), calculated from (segments of) trajectories with $\alpha_{//,\perp} \leq 1.1$, using[47]:

$$D_{//,\perp} = (r_{//,\perp}(t + dt) - r_{//,\perp}(t))^2/dt, \text{ (with } r_{//,\perp} = x, y) \quad (3)$$

averaging locally over 60 displacements, Fig. 4c. We made sure that $D_{\perp}$ was determined for OCR-2 only when it was not actively transported by IFT by adding as a constraint that only time windows were included with $\alpha_{//} < 1.4$, to allow for a proper comparison of the diffusion coefficients in both directions. Using this approach, we found that $D_{//}$ was rather constant, ~0.03 μm²/s, along the cilium, except for the TZ, where it was ~0.01 μm²/s. $D_{\perp}$ was almost three times smaller than $D_{//}$ ~0.01 μm²/s, decreasing somewhat in the distal segment. In part, the lower $D_{\perp}$ compared to $D_{//}$ could be a consequence of our imaging approach, which effectively results in a 2-dimensional projection of a 3-dimensional object, making us blind to motion in the direction perpendicular to the image plane (which is one of the two axes perpendicular to the

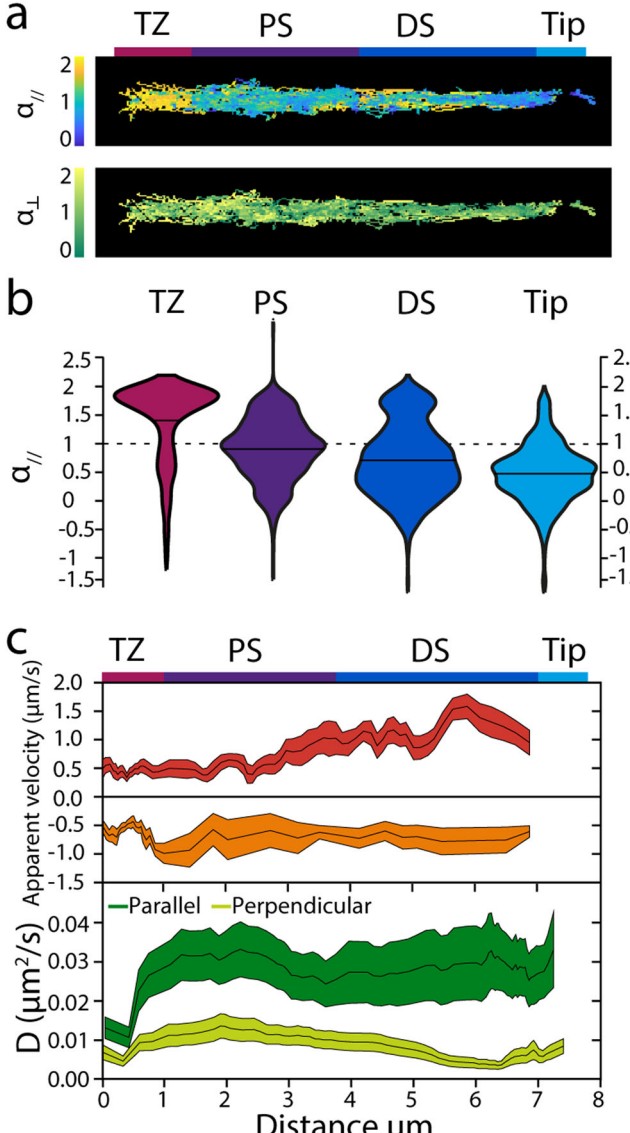

**Fig. 4 Analysis of single-molecule OCR-2 trajectories reveals location-specific motility. a** Heat maps showing all data points color coded for $\alpha_{//}$ and $\alpha_{\perp}$. Scale **b**ar: 1 μm. **b** Apparent velocities and diffusion coefficients of OCR-2 molecules in the phasmid cilia. Top: apparent anterograde and retrograde velocities determined for time points for which $\alpha_{//} \geq 1.4$, averaged over 10 displacements. Line thickness represents s.e.m. Bottom: parallel and perpendicular diffusion coefficients of OCR-2 determined for time points for which $\alpha_{//,\perp} < 1$ averaged locally over 60 displacements. Line thickness represents s.e.m. **c** Violin plots showing $\alpha_{//}$ for the TZ, PS, DS, and Tip regions of the *C. elegans* phasmid cilia.

moving window used in the MSD analysis of the experimental trajectories. Figure 5b shows that the simulated steady-state distribution of OCR-2 reproduces very well the presence of the 3 peaks observed experimentally: at the PCMC, in the beginning of the PS, and at the tip. We note that we can only qualitatively compare the simulations to the experimental data, since in the simulations we have modeled the cilium as a one-dimensional object, disregarding variations in diameter (and consequently membrane area) along the length and the shape of the PCMC. Remarkably, simulations with active transport switched off, reproduce qualitatively the steady-state distributions of OCR-2 in the bbs-8 mutant background (Fig. 5b compared to Fig. 1g) in the sense that the peaks at the beginning of the PS and tip have disappeared. Use of different input parameters (ratios between diffusion and IFT and/or diffusion coefficients) than we extracted from the experiments results in steady-state distributions that are not in correspondence with the experimentally observed distributions (Supplementary Fig. 5). An interesting aspect of these simulations is that we did not explicitly include the TZ as a diffusion barrier. We did this implicitly, by using the experimentally obtained parameters. In particular, the much higher ratio between IFT and diffusion and the lower diffusion coefficient in the TZ compared to elsewhere cause the dip in the OCR-2 distribution observed experimentally and in the simulations. Together, the simulations confirm that location-specific motility parameters (velocities, diffusion coefficients, and ratio between diffusion and IFT) govern the ciliary distribution of OCR-2.

## Discussion

In this study, we employed single-particle imaging in the chemosensory cilia of live *C. elegans* to show that the transmembrane calcium channel OCR-2 displays complex location-specific motility, ranging from subdiffusion to active transport by IFT. Together, these diverse motility modes help to establish OCR-2 distribution necessary for its function in signal transduction, as further confirmed in computer simulations.

Despite fundamental differences in ciliary architecture, our findings on the motility of OCR-2 in *C. elegans* phasmid cilia qualitatively agree with those of TPs in primary cilia of cultured mammalian cells. In *C. elegans*, we found that OCR-2 motility is mostly due to diffusion and only less than 20% by IFT. Also in mammalian primary cilia, GPCRs have been found to diffuse, and to be transported only partly by IFT (~32–34%[33] and ~1%[31] IFT for Smoothened; ~13–27% for SSTR3[33]; ~4% for Patched1[32]). Remarkably, diffusion coefficients obtained for mammalian cilia are substantially larger (SSTR3: ~0.25 μm²/s using FRAP[33]; Smoothened ~0.26 μm²/s, using single-molecule tracking[31]; PTCH1: ~0.1 μm²/s, PTCH1 using single-molecule tracking[32]) than what we observed in *C. elegans* phasmid cilia (~0.03 μm²/s). It is unclear whether these are actual differences or caused by the differences in measurement and analysis approaches, which could differ in their ability to discriminate active transport from diffusion. A lower diffusion coefficient could also be caused by the larger size of the transmembrane section of the calcium channel OCR-2 compared to the mammalian GPCRs, or by species-specific differences in ciliary membrane viscosity, potentially caused by protein crowding[50]. Notwithstanding these qualitative differences, our results and those in mammalian primary cilia indicate that the combination of diffusive and active transport motility modes of TPs, and their ability to connect and disconnect to underlying IFT trains, is a conserved mechanism for the motility of TPs in cilia. More than in these previous studies on mammalian primary cilia, our studies in *C. elegans* phasmid cilia, which are longer and have structurally different microtubule domains, have revealed that TP motility in cilia is segment dependent: in the TZ, OCR-2 is mostly transported directionally

cilium). This could result in an underestimation of the diffusion coefficient in the direction perpendicular to the cilium, at most by only a factor of about 2[48,49]. We speculate that the larger difference between the apparent $D_{//}$ and $D_{\perp}$ in DS and Tip might be caused by confinement, which could result in subdiffusive motion.

To test whether the ciliary distribution of OCR-2 (Fig. 1e) can indeed be reproduced from the motility parameters extracted from the single-molecule trajectories, we performed computer simulations (Fig. 5). We allow OCR-2 to move along an in silico cilium, either by IFT or diffusion, with ratios, velocities and diffusion coefficients extracted from the experimental trajectories (Fig. 5a). We keep motility mode and parameters constant for the duration of the

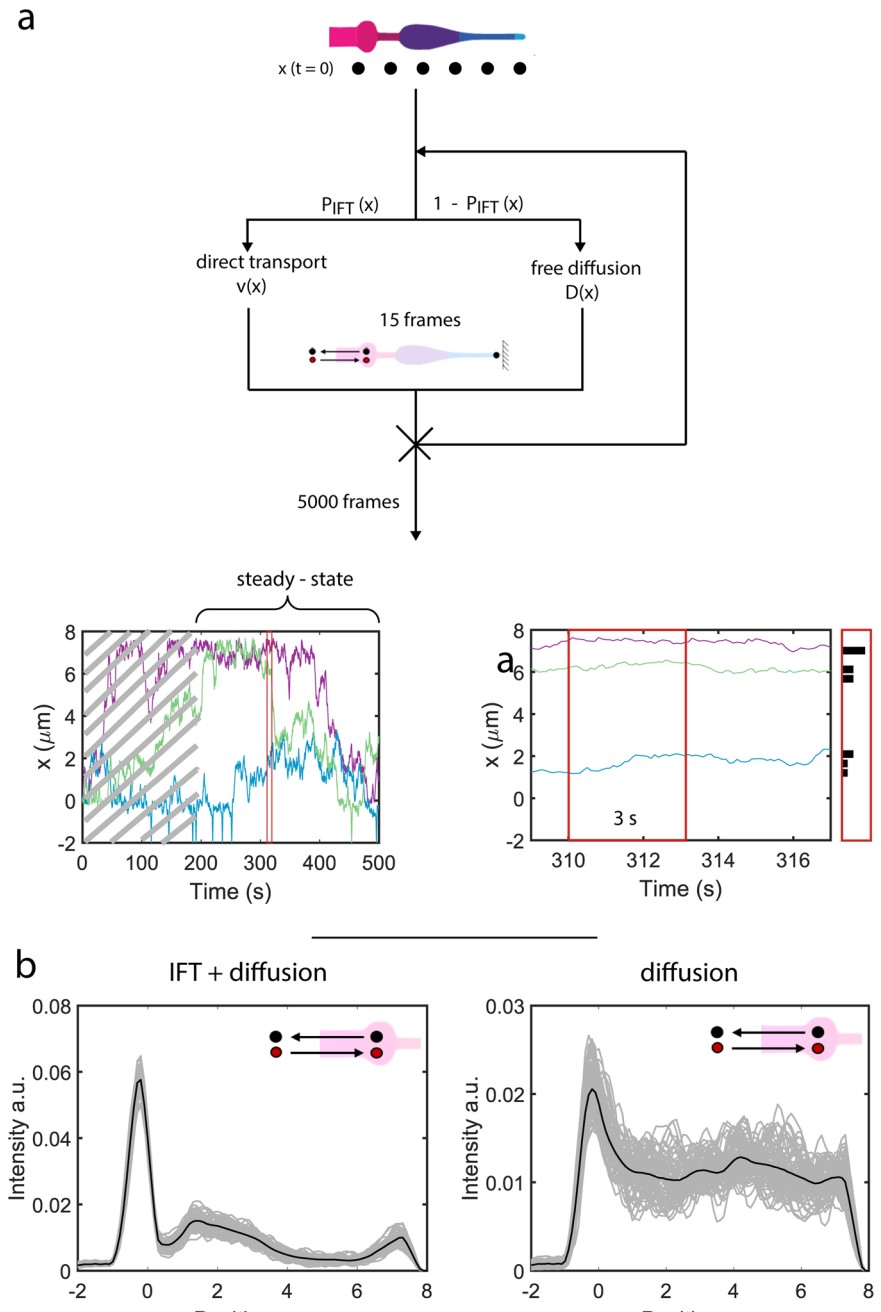

**Fig. 5 Computer simulations of OCR-2 intensity profiles. a** Cartoon depicting the simulations (see "Methods" section) and resulting trajectories (bottom). After reaching steady state (after ~180 s in trajectory bottom left) the positions of the molecules are collected within a time window of 3 s (zoom of trajectory including position histogram, bottom right). **b** Simulated OCR-2 intensity profiles with the velocities and diffusion coefficients extracted from the analysis of the single-molecule trajectories as input parameters (Fig. 4d and Supplementary Table 1). Left: IFT/diffusion ratios used correspond to the values in Supplementary Table 1. Right: the IFT/diffusion ratio is set to 0, in order to mimic OCR-2 in the bbs-8 mutant background. The number of OCR-2 is constant and each OCR-2 leaving the cilium towards the dendrite is replaced by one entering. Gray lines represent the averaged distributions of OCR-2 over a time laps of 3 s (similar to the experimental data in Fig. 1a), after reaching steady state. Black lines represent the distribution averaged over the complete steady-state regime (between 180 and 390 s (see also trajectories in **a**; i.e., all gray curves averaged). Cartoon insets indicate that in the simulations the number of OCR-2 was forced to be constant (in time).

by IFT; in the PS, normal diffusion is prevalent; in the DS, sub-diffusion is most common, with noticeably more directed transport than in the PS; in the tip, OCR-2 motility is rather low, mostly subdiffusive. These location-dependent motility modes and parameters underlie the steady-state distribution of OCR-2 along the phasmid cilia.

What would be the molecular basis for this location-dependent motility? Most likely there are two distinct processes underlying these observations. The first process is a location-dependent connection of OCR-2 with the IFT machinery. The interaction of OCR-2 and other TPs to IFT trains has been shown to be mediated by the BBSome protein complex and by Tubby proteins,

which are essential for chemotaxis[12,13,34]. Recent cryo-EM studies of the BBSome have provided important insights into the molecular details of this interaction: a negatively charged cleft on the BBSome surface that is likely involved in the binding of cargo proteins[51]. In another study, it has been suggested that the GTPase ARL6/BBS-3 is activated when it interacts specifically with the BBSome, allowing for the connection between IFT-trains and transmembrane cargoes[52]. This agrees with our results, which suggest that this interaction is tightly regulated, being persistent in the TZ, and far more intermittent in the rest of the cilium. Furthermore, it has been shown that the distribution of OCR-2 changes after stimulation of the *C. elegans* phasmid chemosensory cilia with adversive chemicals and hyperosmotic solutions[35]. Taken together with our current findings, this strongly suggests that the interaction of OCR-2 with the IFT machinery is not only location dependent, but also affected by sensory activity in the cilium. Further studies will be required to better understand the interaction of TPs and the IFT machinery and how this is regulated.

The second process underlying the location-dependent motility is the location-dependent (sub)diffusion. We have shown that OCR-2, when not connected to the IFT machinery, diffuses normally (with $\alpha \approx 1$) in the PS, while it moves subdiffusively in the DS and Tip (with $\alpha \ll 1$). Most likely, subdiffusion in the parallel direction is caused by an increased density of OCR-2 or other TPs (crowding), and/or static structural proteins, all hampering the motion of OCR-2 in the membrane. In addition, differences in membrane composition and curvature could also play a role, especially in the perpendicular direction. To get a deeper insight into the molecular details of OCR-2 motility, it might be very interesting to increase the imaging frame to >100 frames per second, approaching the stepping rate of the IFT motors, for example by making OCR-2 fusion proteins with more than one GFP and fast imaging approaches[53].

In conclusion, we have shown that sensitive fluorescence microscopy is an invaluable tool to study single-protein dynamics inside living multicellular organisms. Investigating the molecular machinery underlying the development and function of cilia provides important insights into mechanisms involved in key life processes such as sensing, signal transduction, limb patterning, and human disorders collectively named ciliopathies[54,55].

## Methods

***C. elegans* strains**. Strains used in this study were generated using MOSCI insertions and CRISPR/Cas9 genome editing and are listed in Supplementary Table 2[56,57]. Primers and guide RNAs used to make the CRISPR/Cas9 and extrachromosomal array strains are listed in Supplementary Table 3. Maintenance of *C. elegans* was performed using standard procedures[58].

**Fluorescence microscopy**. *C. elegans* young adult hermaphrodites were immobilized using 5 mM levamisole and mounted between a microscope slide with a 2% (W/V) agarose pad, and a coverslip. The worms were subsequently imaged on our custom-build wide-field epifluorescence microscope with 100 ms exposure time, and gently photo bleached to obtain single-molecule sensitivity[59].

**Data analysis**. Kymographs were generated from the image sequences using the open-source ImageJ plugin KymographClear and analyzed using the stand-alone program KymographDirect[60]. The latter was used to determine the velocities of Fig. 2c. Single-molecule tracking was performed using custom-written MATLAB (MathWorks) routines using a linking algorithm[20,61]. To be able to define and distinguish movement parallel and perpendicular to the ciliary axoneme, trajectory coordinates were transposed to a hand-drawn spline using the custom-written MATLAB script. In order to distinguish the anterograde and retrograde velocities of OCR-2, we calculated the sign of the difference between two consecutive parallel positions. If the sign is positive, the directed motion is anterograde.

The super-resolution image for Fig. 1 was generated as before[20,39]. In short, the OCR-2::EGFP signal was gently bleached till visually separable foci could be discerned. After recording a multitude of frames with separable OCR-2::EGFP foci, these were then fitted with a two-dimensional Gaussian to determine the location

of the molecules. A reconstructed image could then be produced from the localizations.

**Fluorescence recovery after photobleaching**. Imaging was performed as described above. For the fluorescence recovery after photobleaching (FRAP) experiments, a region of interest was bleached using a 405 nm laser (Coherent Cube 405 nm 100 mW). The average fluorescence intensity of the same ROI, but next to the cilia, accounting for the background signal, was subtracted from the average fluorescence intensity coming from ROI. FRAP curves were subsequently plotted from the averaged normalized curves of individual worms.

**Computer simulations of the OCR-2 distribution**. For the in silico cilium, a cylinder is used with a constant diameter of 200 nm. The cilium has been divided in different sections along the x-axis: dendrite ($x < -0.8$ μm), PCMC ($-0.8$ μm $\leq x <$ 0 μm), TZ ($0 \leq x < 0.8$ μm), PS ($0.8$ μm $< x < 3.5$ μm), DS ($3.5$ μm $\leq x < 6.5$ μm), and tip ($6.5$ μm $\leq x < 7.5$ μm). The percentage of active transport in the dendrite is set to 100% with an absolute velocity of 1.5 μm/s. For the initial simulation in Fig. 5b, the homogeneous diffusion coefficient ($D$) in the PCMC is set to the $D$ at $x = 0$ μm shown in Fig. 4c. In case OCR-2 is linked to IFT, the probabilities for anterograde- and retrograde-directed transport are set at 1/3 and 2/3, respectively, in order to account for the difference in train frequencies in anterograde and retrograde directions[24]. The mode of transport (directed or diffusive) was randomly chosen on basis of the ratio extracted from the single-molecule analysis. The transport mode is kept constant over a time window of 15 time points (corresponding to 1.5 s), as used in the analysis of the experimental data. At $t = 0$ s, 50 OCR-2 are positioned at each micrometer along the cilium, from 0 to 7 μm. The number of steps in each trajectory is equal to 5000. The averaged intensity profiles are generated by calculating the density of molecules (histogram) over 3 s identical to the intensity profiles generated from the experimental data. The profiles shown are taken from the simulations after a steady-state regime was reached.

**Statistics and reproducibility**. Experiments were performed over several months, and trajectories were recorded in different *C. elegans* specimens. Sample sizes are mentioned in the text and/or figure caption as are the number of worms, trajectories, or individual data points.

**Reporting summary**. Further information on research design is available in the Nature Research Reporting Summary linked to this article.

## Data availability

The datasets of this study are available on DataverseNL: https://doi.org/10.34894/P2GVHU. The scripts used for MSD analysis and modeling are available on GitHub: https://github.com/nbdan01/OCR-2-MSD-analysis.

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

## Acknowledgements

We thank J. Mijalkovic for helpful discussion and support, and Laurent Cognet and Antoine Godin (University of Bordeaux) for help and discussions regarding the analysis of single-molecule trajectories. We acknowledge financial support from the Netherlands Organisation for Scientific Research (NWO) via a Foundation for Fundamental Research on Matter (FOM) program grant ("The Signal is the Noise") and from the European Research Council under the European Union's Horizon 2020 research and innovation program (Grant agreement no. 788363; "HITSCIL").

## Author contributions

Conceptualization and methodology, J.v.K and E.J.G.P.; investigation, JvK; formal analysis, J.v.K. and N.D.; resources, J.v.K. and E.J.G.P.; writing—original draft, J.v.K. and E.J.G.P.; writing—review and editing, J.v.K., N.D., and E.J.G.P.; funding acquisition and supervision, E.J.G.P.

## Competing interests

The authors declare no competing interests.
