## [Peer Review File · Communications Biology]

Reviewers' comments:

Reviewer #1 (Remarks to the Author):

In this manuscript, the authors utilized quantitative fluorescence microscopy approaches to study dynamics underlying the distribution of the sensory trans-membrane cargo OCR-2 in *C.elegans*. Using single molecule imaging of the OCR-2 trajectories the authors show that OCR-2 movement differs in specific cilia sub-locations, being either coupled to IFT, diffusive, or even sub-diffusive.

Their results indicate that the IFT-driven transport of OCR-2 seems to play a major role during crossing the transition zone. In contrast, within cilia bi-modal distribution of motility, such as IFT-coupled and saltatory movement, is observed. Further, their quantitative analysis of the OCR-2 trajectories and FRAP measurements suggests that IFT-directed transport has a greater role in the distal part of the cilium.

Finally, the authors extracted the motility parameters and used them to model computationally OCR-2 distribution within cilia. The simulations supported the notion that diffusion and IFT-directed transport are involved in this process, albeit did not really mimic the experimental data.

Overall, the manuscript reports very interesting results, which in principle advance our understanding of cilia receptor dynamics and localization in cilia. The data are robust and would be of interest to the broader readership after addressing the points enlisted below.

Major points:

1. The authors propose that the location-specific motility modes of the IFT cargo will improve our understanding of the IFT dynamics and the cilia function. According to the data provided, the OCR-2 distribution as such seems to be a result of IFT, kinesin II and OSM-3 cooperation during the steady state. The authors should however also provide data showing whether or how this location – specific motility of OCR-2 changes upon stimulation. Since the OCR-2 distribution is slightly different in the *kap-1* mutant but this mutant is still functional, it will be beneficial to see how the motility modes change in both backgrounds and correlate it with the functional output.

2. Followup of the 1. point. Are there any mutations in the OCR-2 calcium channel that could be tested whether and how they affect its motility and then the functional output of this receptor?

3. The plateaus of the FRAP recovery curves seem to decline, which usually happens due to photobleaching. It is therefore possible that the authors underestimate the amount of the mobile fraction. Did the authors check for this? Are the differences between the 18% and 21% actually significant, what was the statistical analysis used here?

Authors should also mention that the diffusional barrier at the transition zone might limit the FRAP recovery as well, as shown before in Hu et al. 2010. Since the immobile fraction of OCR-2 makes up to 80% of the ciliary signal according to their data, the authors should discuss its contribution to the OCR-2 distribution and function as well in the discussion part.

4. It is not clear, which trajectories were used for quantification of the single-molecule trajectories. The authors show that the velocities of OCR:2 were calculated from 51 anterograde trajectories and 273 retrograde ones in F2C. On the other hand, they say that they used all 185 trajectories in the unbiased analysis in F3E. Could the authors clarify which trajectories (anterograde/retrograde) were used for the individual types of analysis?

5. The authors should provide the information about data visualization and statistical analysis used in F2C. The authors claim that there is no difference between the kinesin II and OCR-2 velocity, however it is not clear from the way the data are presented.

6. The computational simulations do not really mimic the experimental data suggesting that either the authors simplified the analysis of the complex OCR2 motility too much or there are additional factors involved. Changes of the parameters helped a bit in particular locations of cilia, so it is possible that

the final distribution results indeed from more complex interplay of the IFT and diffusion. Could the authors discuss this more? Would it be possible to include additional factors into the predictions, e.g. diffusional barrier?

Minor points:

1. Figure S1: panels A and B are not self-explanatory in contrast to description of Figure 1A, F, etc. Please also explain those 3 different pictures in the S1B, maybe by adding a cilia scheme to it as in other panels.

Typo in legend : salutatory should be saltatory

2. Figure S3: legend: referring to diagram in figure 1, but the nice diagram is only in figure 2A. Could the authors put such diagram rather already to figure 1?

MAIN TEXT with Figures

3. Page 4, abbreviations DS, PS not explained.

4. Page 5, line 8: It is the upper image in Fig. 1g that shows the superb OCR accumulation in PCMC, not the bottom. Please correct.

5. Page 6, line 12: extra word..OF, we generated....

6. Page 6, line 16: rewriteand are osmotic avoidance defective....to: and are defective of osmotic avoidance,

7. Page 6, line 32, extent of

8. Page 6, line 38, segments IN wild type

9. Page 9, line 2...kymographs of Figure 2A

10. Page 10, line 11...values of WHAT can be.....correct

11. Page 10, line 22....were color coded for WHAT and plotted...correct

12. Page 10, Figure 4, change the panel C for B and panel B for C as are the links in the text

13. Page 12, line 1 and 2...too many remarkably

Reviewer #2 (Remarks to the Author):

The Peterman lab reported their single-particle imaging and tracking experiments to examine the motility and dynamics of TRPV-channel OCR-2 along worm sensory cilia. The behavior of TRPV channel on the plasma membrane is a fundamental and fascinating problem in cell biology and neuroscience. The dynamics of OCR-2 was reported by the Rosenbaum lab many years ago; however, the imaging methodology has been significantly improved during the past decade. The Peterman lab is a leading expert on this topic and has used the similar experimental pipeline to make several contributions to the field. This study focused on OCR-2, and the experiments are well designed and performed. The results support their major conclusion. Overall, it is an excellent paper and should be published in Communications Biology with minor text revisions;

1. The abstract can be expanded a little bit. The first three sentences can be shortened, but the ones summarize the key findings should be expanded. For example, the authors can provide more specific information where OCR-2 moves, diffuses, or immotile.

2. Many of their results should be discussed in the light of Xie et al. EMBO, J, 2020. Xie et al reported the behavior of IFT-motor and IFT-particles at >200 Hz imaging rate and discovered the motor pausing for more than 70% imaging time. OCR-2 pausing or other behavior can be explained by the property of IFT system.

Reviewer #3 (Remarks to the Author):

The paper by van Krugten, Danne, and Peterman looks at the interplay between intraflagellar transport (IFT) and diffusion (normal and subdiffusion) to affect the distribution of transmembrane proteins using OCR-2.

It was previously shown that the ion channel OCR-2 exhibits diffusive motion and is also an IFT cargo. Here, the authors perform single molecule imaging to study the movement of OCR-2 in detail. The authors generated two worm lines expressing egfp tagged transmembrane proteins. One expressing SRB-6:eGFP (a GPCR protein) had low expression level and could not be used for single molecule imaging. The second OCR-2:eGFP had a much higher expression level and was used for this study.

They studied ensemble distributions, used kymographs to look at particle movement and FRAP studies in wild type and BBS-8 and kap-1 mutant worms. They find that both the distribution of OCR-2 and the mode of transport varies along the cilia. Results are quantitative. Finally to test their quantitative results they used computer simulations to recreate the observed ciliary distributions from the motility parameters extracted from the single molecule measurements.

To my knowledge, these are novel findings. The data is solid and convincing. This paper moves the field forward, adding information about specific cargo molecules to the detailed information about IFT motors previously elucidated.

Suggested changes:

- 1) Define all abbreviations.
- 2) Define base in Figure 1 with respect to cilia drawing in figure 2

Reviewer 1

In this manuscript, the authors utilized quantitative fluorescence microscopy approaches to study dynamics underlying the distribution of the sensory transmembrane cargo OCR-2 in *C. elegans*. Using single molecule imaging of the OCR-2 trajectories the authors show that OCR-2 movement differs in specific cilia sub-locations, being either coupled to IFT, diffusive, or even sub-diffusive. Their results indicate that the IFT-driven transport of OCR-2 seems to play a major role during crossing the transition zone. In contrast, within cilia bi-modal distribution of motility, such as IFT-coupled and saltatory movement, is observed. Further, their quantitative analysis of the OCR-2 trajectories and FRAP measurements suggests that IFT-directed transport has a greater role in the distal part of the cilium. Finally, the authors extracted the motility parameters and used them to model computationally OCR-2 distribution within cilia. The simulations supported the notion that diffusion and IFT-directed transport are involved in this process, albeit did not really mimic the experimental data. Overall, the manuscript reports very interesting results, which in principle advance our understanding of cilia receptor dynamics and localization in cilia. The data are robust and would be of interest to the broader readership after addressing the points enlisted below.

We thank the reviewer for their summary of our findings and the positive overall evaluation of the manuscript.

Major points:

1. The authors propose that the location-specific motility modes of the IFT cargo will improve our understanding of the IFT dynamics and the cilia function. According to the data provided, the OCR-2 distribution as such seems to be a result of IFT, kinesin II and OSM-3 cooperation during the steady state. The authors should however also provide data showing whether or how this location – specific motility of OCR-2 changes upon stimulation. Since the OCR-2 distribution is slightly

different in the kap-1 mutant but this mutant is still functional, it will be beneficial to see how the motility modes change in both backgrounds and correlate it with the functional output.

This is a very interesting remark by the reviewer. We have recently finalized a detailed study of the effect of chemical stimulation on IFT, and ciliary structure and components using microfluidics. In this study, which can be found on BioArxiv (Bruggeman et al. 2022). we indeed show that OCR-2 (and also many other ciliary components) redistribute upon stimulation with adversary chemicals and hyperosmotic solutions. This strongly suggests that the motility mode of OCR-2 is not only location-specific but also affected by stimulation of the cilia. We have added a reference to that work and discussed its implications for the current study.

2. Follow-up of the 1. point. Are there any mutations in the OCR-2 calcium channel that could be tested whether and how they affect its motility and then the functional output of this receptor?

The reviewer again raises a very interesting point. Unfortunately, we are not aware of anything known of such OCR-2 mutations. A detailed study of those mutations would require substantial effort and time, outside the scope of the current study / manuscript. This could be a very interesting direction of future research.

3. The plateaus of the FRAP recovery curves seem to decline, which usually happens due to photobleaching. It is therefore possible that the authors underestimate the amount of the mobile fraction. Did the authors check for this? Are the differences between the 18% and 21% actually significant, what was the statistical analysis used here?

Indeed, long time-scale photobleaching might play a role in the FRAP recovery curves. This is exactly why we use the FRAP data only as an initial, qualitative indication of the mobility of OCR-2. We cannot claim that a difference between 18 and 21% is statistically significant. What we extract from the FRAP data is a qualitative indication of the existence of mobile and non-mobile fractions. For quantitative and more detailed insights in the dynamics of OCR-2 we use our single-molecule experiments, further in the manuscript. We have clarified this in the revised version of the manuscript.

Authors should also mention that the diffusional barrier at the transition zone might limit the FRAP recovery as well, as shown before in Hu et al. 2010. Since the immobile fraction of OCR-2 makes up to 80% of the ciliary signal according to their data, the authors should discuss its contribution to the OCR-2 distribution and function as well in the discussion part.

As written above, we intended the FRAP data only as a qualitative indication of the existence of fractions with different motility. For a quantitative discussion we have restricted ourselves to the far more detailed single-molecule data, which we have analyzed and discussed in great detail. Our single-molecule data indeed shows clear evidence of the transition zone acting as a diffusion barrier for OCR-2. We observe a low number of OCR-2 complexes in the transition zone, most of which are not freely diffusing but actively transported. We have changed the text to clarify.

4. It is not clear, which trajectories were used for quantification of the single-molecule trajectories. The authors show that the velocities of OCR:2 were calculated from 51 anterograde trajectories and 273 retrograde ones in F2C. On the other hand, they say that they used all 185 trajectories in the unbiased analysis in F3E. Could the authors clarify which trajectories (anterograde/retrograde) were used for the individual types of analysis?

The reviewer is correct: the trajectories used for Figures 2 and 3 are not the same. For Figure 2C, we optimized experimental conditions to observe transition-zone crossings. The analysis of these 51+273 tracks was performed using the ImageJ plugin KymographClear and KymographDirect, tools we have developed in the lab for this purpose. Velocities were extracted from the slopes in the kymographs (position vs time). The data for Figure 3 is from another dataset, for which experimental conditions were optimized to track single molecules in the entire cilium. In this way, we obtained 185 trajectories that were analysed with single-particle tracking software, and were used for the detailed MSD analysis. We have clarified this in the manuscript.

5. The authors should provide the information about data visualization and statistical analysis used in F2C. The authors claim that there is no difference between the kinesin II and OCR-2 velocity, however it is not clear from the way the data are presented.

As indicated above, these data were obtained from kymographs, using KymographClear and KymographDirect. Data are represented in boxplots, showing the relevant statistical parameters: median, 25-75 and 5-95 percentiles and sample size. We apologize that this was not clearly indicated in the original manuscript. This has been corrected.

We agree that the text (that the velocities are 'similar') is rather handwaving compared to the data in the figure. We have changed the wording to more accurately describe the data.

6. The computational simulations do not really mimic the experimental data suggesting that either the authors simplified the analysis of the complex OCR2 motility too much or there are additional factors involved. Changes of the parameters helped a bit in particular locations of cilia, so it is possible that the final distribution results indeed from more complex interplay of the IFT and diffusion. Could the authors discuss this more? Would it be possible to include additional factors into the predictions, e.g. diffusional barrier?

We thank the reviewer for seeing the value of computational simulations. It has been our explicit intention to keep the simulations as simple as possible. In our view, the simulations serve to see whether the key aspects of the experimentally observed distributions (accumulations at tip, just before and just behind transition zone) can be reproduced from the motility parameters extracted from the single-molecule trajectories. And they are reproduced. Quantitatively, there are differences between simulations and experimental data and these are to be expected, because many important aspects were not taken into account, such as for example the local circumference of the ciliary membrane or the effective depth of field of our microscope images. In our view, the simulations show qualitative similarity with the ciliary

distribution of OCR-2 in the presence and absence of IFT. This confirms that we are not missing major additional factors that affect the motility of OCR-2. We have clarified this in the text of the revised manuscript by substantially extending the discussion of the simulations. Inspired by the reviewer's comment on the effect of the transition zone as a diffusion barrier, we have included a discussion of the transition zone as diffusion barrier in relation to the simulations. In the simulations we do not explicitly take into account the transition zone as a diffusion barrier. We do this implicitly, by using the experimentally obtained parameters. In particular, the much higher ratio between IFT and diffusion and the lower diffusion coefficient in the TZ compared to elsewhere cause the dip in the OCR-2 distribution observed experimentally and in the simulations and make the TZ a diffusion barrier.

Minor points:

1. Figure S1: panels A and B are not self-explanatory in contrast to description of Figure 1A, F, etc. Please also explain those 3 different pictures in the S1B, maybe by adding a cilia scheme to it as in other panels.

We have changed the figures and captions as suggested by the reviewer.

Typo in legend : salutatory should be saltatory

Changed.

2. Figure S3: legend: referring to diagram in figure 1, but the nice diagram is only in figure 2A. Could the authors put such diagram rather already to figure 1?

MAIN TEXT with Figures

We have followed the reviewer's suggestion and have made the annotation/diagram similar for all figures.

3. Page 4, abbreviations DS, PS not explained.

4. Page 5, line 8: It is the upper image in Fig. 1g that shows the superb OCR accumulation in PCMC, not the bottom. Please correct.

5. Page 6, line 12: extra word..OF, we generated....

6. Page 6, line 16: rewriteand are osmotic avoidance defective....to: and are defective of osmotic avoidance,

7. Page 6, line 32, extent of

8. Page 6, line 38, segments IN wild type

9. Page 9, line 2...kymographs of Figure 2A

10. Page 10, line 11...values of WHAT can be.....correct

11. Page 10, line 22...were color coded for WHAT and plotted...correct

12. Page 10, Figure 4, change the panel C for B and panel B for C as are the links in the text

13. Page 12, line 1 and 2...too many remarkably

We thank the reviewer for these remarks that help us to improve clarity and readability of the manuscript. We have addressed all the points in the revised version of the manuscript.

Reviewer 2

The Peterman lab reported their single-particle imaging and tracking experiments to examine the motility and dynamics of TRPV-channel OCR-2 along worm sensory cilia. The behavior of TRPV channel on the plasma membrane is a fundamental and fascinating problem in cell biology and neuroscience. The dynamics of OCR-2 was reported by the Rosenbaum lab many years ago; however, the imaging methodology has been significantly improved during the past decade. The Peterman lab is a leading expert on this topic and has used the similar experimental pipeline to make several contributions to the field. This study focused on OCR-2, and the experiments are well designed and performed. The results support their major conclusion. Overall, it is an excellent paper and should be published in *Communications Biology* with minor text revisions;

We thank the reviewer for their summary of our findings, the appreciation of the importance of our work, and the positive overall evaluation of the manuscript.

1. The abstract can be expanded a little bit. The first three sentences can be shortened, but the ones summarize the key findings should be expanded. For example, the authors can provide more specific information where OCR-2 moves, diffuses, or immotile.

We have followed the reviewer's recommendation and changed the abstract accordingly.

2. Many of their results should be discussed in the light of Xie et al. *EMBO, J, 2020*. Xie et al reported the behavior of IFT-motor and IFT-particles at >200 Hz imaging rate and discovered the motor pausing for more than 70% imaging time. OCR-2 pausing or other behavior can be explained by the property of IFT system.

We thank the reviewer for pointing us to this paper. We note that the time scale of our measurements and those of Xie et al. are quite different (more than an order of magnitude). The time scale of the Xie experiments is comparable to the stepping rate of the motor proteins driving IFT. Very likely, in our (slower) imaging most of the pauses observed by Xie are averaged out. In our current study, we have optimized the imaging conditions to find the right compromise between as long as possible trajectories, single-to-noise and imaging fast enough to accurately extract parameters like velocity and diffusion coefficients (on the 100s of ms time scale) using a minimally perturbing fluorescent probe. It could indeed be very interesting to follow the dynamics of OCR-2 at a much higher time resolution using an approach like that of Xie et al. We have added a reference and a short discussion.

Reviewer 3

The paper by van Krugten, Danne, and Peterman looks at the interplay between intraflagellar transport (IFT) and diffusion (normal and subdiffusion) to affect the distribution of transmembrane proteins using OCR-2. It was previously shown that the ion channel OCR-2 exhibits diffusive motion and is also an IFT cargo. Here, the authors perform single molecule imaging to study the movement of OCR-2 in detail. The authors generated two worm lines expressing egfp tagged transmembrane proteins. One expressing SRB-6:eGFP (a GPCR protein) had low expression level and

could not be used for single molecule imaging. The second OCR-2:eGFP had a much higher expression level and was used for this study. They studied ensemble distributions, used kymographs to look at particle movement and FRAP studies in wild type and BBS-8 and kap-1 mutant worms. They find that both the distribution of OCR-2 and the mode of transport varies along the cilia. Results are quantitative. Finally to test their quantitative results they used computer simulations to recreate the observed ciliary distributions from the motility parameters extracted from the single molecule measurements. To my knowledge, these are novel findings. The data is solid and convincing. This paper moves the field forward, adding information about specific cargo molecules to the detailed information about IFT motors previously elucidated.

We thank the reviewer for their summary of our findings, the appreciation of the data, and the positive overall evaluation of the manuscript.

Suggested changes:

1) Define all abbreviations.

We have followed the recommendation of the reviewer and defined all abbreviations in the manuscript.

2) Define base in Figure 1 with respect to cilia drawing in figure 2

We thank the reviewer for this remark. We have followed the advice and we have improved the annotation in the figures and made all figures consistent in this respect.

REVIEWERS' COMMENTS:

Reviewer #1 (Remarks to the Author):

The revised version of the manuscript is now suitable for publication in COM Biology. The authors clarified very well the concerns and unclear parts of the study both in the experimental and the discussion section.